# Six months survival and risk factors for attrition for patients detected with cryptococcal antigenemia through screening in Malawi

**Master R. O. Chisale**[1,2,3]*, **Alex Jordan**[4], **Pocha S. Kamudumuli**[5], **Bernard Mvula**[6], **Michael Odo**[7], **Alice Maida**[8], **James Kandulu**[9], **Ben Chilima**[10], **Frank W. Sinyiza**[1], **Pauline Katundu**[10], **Hsin-yi Lee**[2], **Rebecca Mtegha**[2], **Tsung-Shu Joseph Wu**[2,11], **Joseph Bitirinyo**[10], **Rose Nyirenda**[7], **Thoko Kalua**[5], **Greg Greene**[4], **Tom Chiller**[4]

1 Ministry of Health, Malawi, Mzuzu Central Hospital, Luwinga, Mzuzu, Malawi, 2 College of Medicine, University of Malawi, Zomba, Malawi, 3 Faculty of Science Technology and Innovations, Biological Sciences Department, Mzuzu University, Mzuzu, Malawi, 4 Centres for Disease Control and Prevention, Mycotic Diseases Branch, Atlanta, GA, United States of America, 5 University of Maryland Global Initiative Corporation (UMGIC), Lilongwe, Malawi, 6 Ministry of Health, National HIV Unit laboratory, Lilongwe, Malawi, 7 Ministry of Health, Department of HIV and AIDS (DHA), Lilongwe, Malawi, 8 Centres for Disease Control and Prevention-Malawi, Malawi, 9 Capitol Hill (Headquarters), Ministry of Health, Ministry of Health, Lilongwe, Malawi, 10 Public Health Institute of Malawi (PHIM), Ministry of Health, Lilongwe, Malawi, 11 Luke International, Mzuzu, Malawi

* masterchisale@gmail.com, chisale.m@mzuni.ac.mw

**Data Availability Statement:** All relevant data are within the paper and its Supporting Information files.

## Abstract

### Main objective

A cohort of adult Malawian people living with HIV (PLHIV) testing positive for cryptococcal antigenemia was observed and followed to determine the outcomes and risk factors for attrition.

### Methods concept

Eligible PLHIV were enrolled at 5 health facilities in Malawi, representing different levels of health care. ART naïve patients, ART defaulters returning to care, and patients with suspected or confirmed ART treatment failure with CD4 <200 cells/µL or clinical stage 3 or 4 were enrolled and received CrAg tests on whole blood specimens from August 2018 to August 2019. Hospitalized PLHIV were enrolled and tested for CrAg from January 2019 to August 2019, regardless of CD4 or clinical stage. Patients with cryptococcal antigenemia were managed per Malawian clinical guidelines and were followed up for six months. Survival and risk factors for attrition at six months were assessed.

### Results

A total of 2146 patients were screened and 112 (5.2%) had cryptococcal antigenemia. Prevalence ranged from 3.8% (Mzuzu Central Hospital) to 25.8% (Jenda Rural Hospital). Of the 112 patients with antigenemia, 33 (29.5%) were diagnosed with concurrent CM at the time

**Funding:** CDC Foundation (https://www.cdcfoundation.org/), USA, is the financier of this project (Project #950). This was funded to Master R.O. Chisale and the funder was not involved in the study in any way. The funders had no role in study design, data collection and analysis, decision to publish, or preparation of the manuscript.

**Competing interests:** The authors have declared that no competing interests exist.

of enrollment. Six-month crude survival of all patients with antigenemia (regardless of CM status) ranged from 52.3% (assuming lost-to-follow-up (LTFU) patients died) to 64.9% (if LTFU survived). Patients who were diagnosed with concurrent CM by CSF test had poor survival (27.3–39.4%). Patients with antigenemia who were not diagnosed with concurrent CM had 71.4% (if LTFU died)– 89.8% (if LTFU survived) survival at six months. In adjusted analyses, patients with cryptococcal antigenemia detected after admission to inpatient care (aHR: 2.56, 1.07–6.15) and patients with concurrent CM at the time of positive antigenemia result (aHR: 2.48, 1.04–5.92) had significantly higher hazard of attrition at six months.

## Conclusions

Overall, our findings indicate a need for routine access to CrAg screening and pre-emptive fluconazole treatment as a way to detect cryptococcal antigenemia and prevent CM in out-patient and inpatient settings. Rapid access to diagnosis and treatment for cryptococcal meningitis (CM) with gold-standard antifungals is needed to improve survival of patients with advanced HIV in Malawi.

## Introduction

Cryptococcal meningitis (CM) is a common opportunistic infection and a leading cause of death among individuals with HIV/AIDS [1]. The annual global burden of disease is estimated to be 152,000 with 112,000 (74%) deaths annually, accounting for an estimated 19% of all AIDS-related mortality. Sub-Saharan Africa is estimated to account for 63% of all CM cases worldwide and cryptococcal infection is the leading cause of meningitis among adults in some parts of sub-Saharan Africa [2]. Despite increased availability of antiretroviral therapy (ART) and anti-fungal therapy, CM continues to be associated with high mortality rates, ranging from 20%-70% in high to low-income countries [3]. The majority of HIV-associated cases have historically been detected in ART-naïve patients, but a growing proportion of cases are now observed in ART-experienced patients [4]. ART-associated cases of CM that occur shortly after initiation of ART are also associated with significant mortality [5].

*Cryptococcus spp.* contain a capsular polysaccharide, known as cryptococcal antigen (CrAg), which can be detected in blood (antigenemia) weeks to months before onset of CM [6]. A targeted "screen and treat" approach using a highly sensitive and specific CrAg lateral flow assay (CrAg LFA) to detect and pre-emptively treat patients with cryptococcal antigenemia (who have not yet developed CM) has shown to reduce the incidence of CM as well as overall AIDS-related mortality [7, 8]. The WHO recommends CrAg screening for PLHIV with CD4 <100 cells/mm$^3$, and for those with CD4<200/ mm$^3$ where resources allow [9]. An increasing number of countries in Sub-Saharan Africa now recommend CrAg screening as part of national HIV care guidelines [10]. A strategic framework which outlines the diagnostic and treatment interventions needed to reduce CM mortality has recently been developed as part of a strategy to End Cryptococcal Meningitis Deaths by 2030 [11]

In 2018, the Malawi department of HIV/AIDS and Ministry of Health released national HIV guidelines which recommended targeted CD4 testing for any ART-naïve PLHIV, PLHIV who had returned from interruption in treatment, and PLHIV with suspected or confirmed treatment failure [12]. CrAg screening was recommended for any PLHIV with a CD4<200, a clinical stage 3–4 illness or admitted to hospital as an inpatient. A pilot of CrAg screening and

pre-emptive fluconazole treatment was conducted to assess the prevalence of cryptococcal antigenemia and concurrent CM at different levels of healthcare in Malawi. Patients with anti-genemia detected in the study were followed for six months in order to better understand the outcomes and risk factors of attrition (either LTFU or death) at six-months for patients with cryptococcal antigenemia.

## Methods and materials

### Study design and participating sites

This was a prospective observational cohort study which aimed to assess six month outcomes in patients identified with cryptococcal antigenemia during a 12-month pilot of CrAg screening (with enrollment beginning in August 2018 and concluding in February 2020). Participants were enrolled from 5 sites, representing each of the three regions of Malawi and each level of healthcare (primary, secondary, tertiary). The target population mirrored Malawian recommendations and consisted of adult ($\geq$18 years) PLHIV admitted to inpatient departments (IPD), PLHIV who were ART treatment naïve, PLHIV who had interrupted treatment and were returning to care, and PLHIV with suspected or confirmed ART treatment failure (targeted / repeat VL result of $\geq$1,000). Adult PLHIV meeting any of these criteria were assessed for advanced HIV disease (AHD) through CD4 testing and clinical staging. Patients presenting to outpatient departments (OPD) with CD4<200 cells/ mm$^3$ or clinical stage 3–4 illness were enrolled from August 2018-August 2019. Adult PLHIV who were admitted to IPD were enrolled from January 2019 to August 2019. Patients enrolled from OPD received CrAg testing if AHD was detected per CD4 or clinical stage criteria and patients enrolled from IPD received CrAg testing regardless of CD4 result or clinical stage. All patients with detected cryptococcal antigenemia were followed for 6 months from enrolment date. The study excluded patients who had been previously diagnosed or treated for CM, patients who had contraindications for the use of fluconazole, and patients who were pregnant or breastfeeding.

### Data collection procedures

Testing for cryptococcal antigenemia (CrAg) was conducted using the CrAg LFA (Immy Co., Norman, USA) on whole blood specimens gathered by venous or pipetted fingerprick collection, depending on patient condition and preference. All CrAg tests were conducted at the point of care following CD4 testing with PIMA (Abbott., Chicago, USA). All laboratory work was conducted using standardized SOPs and standard diagnostic package inserts [13]. Laboratory and clinical data were captured at the time of enrolment and at follow-up visits (for patients with antigenemia) using laboratory registers and clinical charts developed for the study. Quality controls (both positive and negative) were conducted at each site on a daily basis. Lot-to-lot quality controls were also conducted when new test kits were opened.

### CrAg screening and treatment

All patients detected with cryptococcal antigenemia at the time of enrolment underwent a clinical signs and symptoms screening for CM per Malawian guidelines (i.e., headache, fever, confusion, convulsions, neck stiffness). Patients with antigenemia displaying any signs or symptoms of CM were prioritized for lumbar puncture (LP) to obtain cerebrospinal fluid (CSF) for diagnostic testing. Asymptomatic patients were also asked to receive an LP for CSF CrAg testing, per the 2018 WHO guidelines on management of cryptococcal disease [9]. Patients diagnosed with CM were prescribed 2 weeks induction therapy with amphotericin B deoxycholate and fluconazole per the Malawian HIV program standard of care at the time of

the study. Patients diagnosed with CM at facilities without access to amphotericin B were referred to the closest referral hospital with amphotericin B for CM treatment. Patients with antigenemia who were not diagnosed with CM received pre-emptive fluconazole treatment. Pre-emptive antifungal therapy for patients with antigenemia (and no concurrent CM) followed the Malawian National HIV care recommendation: 800 mg/day of fluconazole for the first two weeks, followed by eight weeks of 400mg of fluconazole daily, and then 200 mg/day fluconazole maintenance dosage. All patients with antigenemia were followed through routine methods including phone calls and were assessed at each subsequent facility visit by the study nurse and treating clinician/medical officer, to assess for any new symptoms of CM and to evaluate adherence and assess potential side effects of antifungal therapy. While ART initiation was recommended to be delayed for all ART-naïve patients diagnosed with CM, Malawi HIV guidelines in place at the time of the pilot did not make an explicit recommendation on timing of ART initiation for ART-naïve patients with cryptococcal antigenemia who were not diagnosed with CM. National guidance also did not make a recommendation related to ART re-initiation or regimen switching based on cryptococcal disease status in ART-experience patients. Therefore, all ART initiation, re-initiation, and regimen switching were done per clinician discretion based on standard of care at the pilot facilities during the study period.

## Data analysis

Data were collected periodically from study sites and uploaded to an excel database and imported to SAS 9.4 (SAS Institute Inc., Cary, NC, USA) for analysis. Descriptive statistics for all demographic and clinical variables were calculated and presented as proportions (categorical), median, and interquartile range (continuous).

Kaplan-Meier survival curve analysis and Cox proportional hazards regression were conducted to identify factors associated with attrition (death or LTFU) at six months for patients with antigenemia. Twenty-eight patients with antigenemia who were either LTFU or known to have died did not have a known period of time in care. Patients who were known to have died but with an unknown date of death had a random date of death generated that fell from 1–7 days following their last known date in care. Patients who never returned to care after enrolment and were LTFU even after tracing attempts had a censoring date generated that fell from 1–7 days from the date of the positive CrAg test. This short date range was chosen because patients with antigenemia who were no longer engaged with care were considered likely to have considerable early mortality. Sensitivity analysis was conducted using a wider range of randomly generated dates to ensure that our choice of date range did not bias our findings. The significance of associations identified in the survival analysis did not change when using the wider range of 1–21 days, or when excluding patients with unknown follow up time completely. Any variable with a p-value ≤0.2 was considered in the initial adjusted Cox Proportional Hazards model. Variables were selected for the final model based on assessment of effect size and precision when removing variables in backward model selection. Cox proportional hazards regression was also conducted separately for patients with antigenemia enrolled from OPD and IPD. An additional adjusted model was run which included only patients with antigenemia who were not diagnosed with concurrent CM.

## IRB statement and disclaimer

The study was conducted according to the guidelines of the Declaration of Helsinki and approved by the National Health Science Research Committee (NHSRC) of Malawi (Protocol # 18/01/1961). CDC staff were determined to be non-engaged in the pilot screening activities and therefore the pilot was determined to be exempt from Centers for Disease Control and

Prevention (CDC) IRB review. The findings and conclusions in this report are those of the authors and do not necessarily represent the official position of the Centers for Disease Control and Prevention (CDC)

## Informed consent

Patient written consent was taken from all the patients as per nature of this study and conditions stipulated in our local research ethics approval body.

## Results

A total of 2146 patients were screened for CrAg during the pilot study, and 112 (5.2%) patients were identified with cryptococcal antigenemia (Table 1). Overall prevalence of antigenemia ranged from 3.8% (Mzuzu Central Hospital) to 25.8% (Jenda Rural Hospital). The prevalence in patients with CD4<200, CD4 101–199, and CD4≤100 was 7.9%, 3.7%, and 11.5%, respectively. Additional results on the prevalence and risk factors for cryptococcal antigenemia will be presented in a separate manuscript. CM symptom assessments were conducted for all patients with antigenemia and revealed that 54.2% displayed at least one symptom of CM at the time of positive CrAg test. Of all patients with antigenemia who were symptomatic for CM, 95% received an LP and subsequent CSF-CrAg test. Of these, 59% were diagnosed with concurrent CM based on positive CSF-CrAg results. Of the patients diagnosed with concurrent CM, 13 (39%) were enrolled from OPD. Of these OPD patients with antigenemia and concurrent CM, 9 had either defaulted from ART and were returning to care or were on ART with suspected or confirmed treatment failure. No asymptomatic patients agreed to receive an LP and CSF CrAg test to exclude subclinical CM. Of the 33 patients detected with concurrent CM at enrolment, almost 85% had a headache at the time of positive CrAg test. Neck stiffness, fever, confusion, and convulsions were present for 49%, 46%, 27%, and 18% of patients with concurrent CM, respectively. The majority (73%) of patients with concurrent CM had more than one of these symptoms at the time of CM diagnosis.

Table 1. Prevalence of cryptococcal antigenemia and six-month survival of patients with antigenemia by facility.

| Facility (level of care) | Total CrAg tested | Cryptococcal antigenemia [a] | Symptomatic for CM (%[b]) | CSF-test received (%[c]) | CSF-positive (%[d]) | LTFU (%[b]) | Known deaths (%) | 6-month survival (assuming LTFU died) (%[b]) | 6-month survival (assuming LTFU alive) (%)[b] |
|---|---|---|---|---|---|---|---|---|---|
| Mzuzu (Tertiary) | 906 | 34 (3.8) | 17 (50) | 15 (88.2) | 11 (73.3) | 6 (17.6) | 13 (38.2) | 14 (42.4) | 20 (60.6) |
| Jenda (Rural Secondary) | 124 | 33 (26.6) | 7 (21.2) | 7 (100) | 3 (42.9) | 4 (12.1) | 9 (27.3) | 20 (60.6) | 24 (72.7) |
| Dedza (Secondary) | 301 | 13 (4.3) | 11 (84.6) | 11 (100) | 4 (36.4) | 0 (-) | 5 (38.5) | 8 (61.5) | 8 (61.5) |
| Bangwe (Primary) | 166 | 10 (6.0) | 4 (40) | 3 (75) | 2 (66.7) | 2 (20) | 1 (10) | 7 (70) | 9 (90) |
| Thyolo (Secondary) | 649 | 22 (3.9) | 20 (90.9) | 20 (100) | 13 (65) | 2 (9.1) | 11 (50) | 9 (40.9) | 11 (50) |
| Totals | 2146 | 112 (5.2) | 59 (52.7) | 56 (94.9) | 33 (58.9) | 14 (12.6) | 39 (35.1) | 58 (52.3) | 72 (64.9) |

Note

[a]One patient who transferred out of Mzuzu to a non-pilot facility prior to 6 months is included in the total, but is excluded from the survival calculations and not considered for the denominator for LTFU, Known deaths, or 6-month survival

[b]Denominator for the percentage is the total with cryptococcal antigenemia

[c]Denominator is the total symptomatic for CM

[d]Denominator for the percentage is the total receiving a CSF CrAg test

Coverage of fluconazole prescribing was very high for patients with antigenemia who were asymptomatic (100%) or symptomatic with negative CSF-CrAg results (96.2%) (**Supplemental material**). Of the patients with concurrent CM, 55% were confirmed to have been prescribed amphotericin B deoxycholate and fluconazole, 8 (24%) were missing treatment data, and 6 (18%) were prescribed fluconazole monotherapy. Of 6 patients with concurrent CM who were known to have been prescribed fluconazole monotherapy, only 1 was known to have survived to 6 months. Twenty-nine ART-naïve patients enrolled from OPD were detected with antigenemia and not diagnosed with concurrent CM. Of these, 25 (86%) were known to have been prescribed ART, 1 (3%) was known to have not been prescribed ART and 4 (14%) had missing ART initiation data. Of the outpatients known to have been prescribed ART 12 (48%) delayed ART initiation due to concerns for immune reconstitution inflammatory syndrome (IRIS) (median time to ART initiation: 15 days, range: 13–34 days).

No patients with antigenemia who were asymptomatic, symptomatic with CSF-negative results, or symptomatic with no CSF test at the time of enrolment were known to be diagnosed with CM during the follow up period. Four patients who were either asymptomatic or CSF-CrAg-negative at baseline did develop symptoms that could have been indicative of CM during follow up (mean time to new symptoms: 36 days) (**Supplemental material**). All four of these patients reported headache and two also reported fever. Of these patients, three saw the new symptoms subsequently resolve and survived to six months. The fourth patient died on the day that symptoms were reported, with cause of death reported as TB and severe bacterial infection. None of these 4 patients received CSF CrAg tests to rule out CM.

Six-month crude survival of all patients with antigenemia ranged from 52.3% (if LTFU died) to 64.9% (if LTFU survived) (**Table 1**). The overall proportion of patients with antigenemia who were known to have survived to six months ranged from 40.9% (Dedza Secondary Hospital) to 70% (Bangwe Primary Clinic) by site. Survival of patients with antigenemia who were asymptomatic at the time of positive CrAg test ranged from 71.4% (if LTFU died) to 89.8% (if LTFU survived). For patients who were diagnosed with concurrent CM based on positive CSF-CrAg result, survival ranged from 27.3% (if LTFU died) to 39.4% (if LTFU alive).

There was no significant difference in six-month known survival (not LTFU or died) by gender, age group, or study site. Overall known survival of patients with antigenemia enrolled from IPD was significantly lower than that of patients with antigenemia enrolled from OPD (log-rank: <0.0001) (**Table 2**). Survival of patients with antigenemia differed significantly by CM symptom/CSF status (i.e. asymptomatic, symptomatic/CSF-, CSF+/CM) (log-rank: <0.0001). In stratified Kaplan-Meier analysis, six-month known survival was significantly associated with CM symptom/CSF status for those enrolled as outpatients (OPD) (log-rank: 0.002) (**Fig 1A**). The 6-month survival in asymptomatic patients with antigenemia enrolled from OPD was 75.6% (if LTFU died)– 95.1% (if LTFU survived). Survival in OPD patients diagnosed with concurrent CM ranged from 30.8% (if LTFU died) to 46.2% (if LTFU survived). Six-month survival of patients with antigenemia enrolled from inpatient departments (IPD) did not differ significantly by CM symptom/CSF status, but was very low overall (23.1% if LTFU died—33.3% if LTFU survived) (**Fig 1B**). Significant differences in six-month survival by baseline CD4 (log rank: 0.02) and baseline clinical stage (log-rank: 0.02) were observed in unadjusted analysis, with six-month survival positively correlated with higher baseline CD4 and lower baseline clinical stage (**Table 2**). No significant difference in outcomes was observed in ART naïve patients without concurrent CM who initiated ART immediately vs delayed.

Simple Cox proportional hazards univariable regression analysis identified significant associations between reason for enrolment (ART naïve, return to treatment, treatment failure, admitted to IPD), CD4 at baseline, clinical stage at enrolment, CM symptom/CSF status and the hazard of attrition (LTFU or death) at six months for patients with antigenemia (**Table 3**).

**Table 2. Clinical characteristics of patients with antigenemia and survival range by LTFU assumption.**

| | Cryptococcal antigenemia (%)[a] | Survived at 6 months (assuming LTFU died) | Survived at 6 months (assuming LTFU survived) | p-value[d] |
|---|---|---|---|---|
| | N = 111 | n = 58 (52.3%) | n = 72 (64.9%) | |
| Location of Enrollment | | | | |
| Outpatient | 72 (64.9) | 49 (68.1) | 59 (81.9) | <0.0001[e] |
| ART naive | 33 (29.7) | 23 (69.7) | 26 (76.5) | <0.0001 |
| Defaulter | 12 (10.8) | 6 (50.0) | 9 (75.0) | |
| ART Failure | 27 (24.3) | 20 (74.1) | 24 (88.9) | |
| Inpatient | 39 (35.1) | 9 (23.1) | 13 (33.3) | |
| CD4 | | | | |
| CD4≥200 | 12 (10.8) | 8 (66.7) | 11 (91.7) | 0.02 |
| CD4 101–199 | 22 (19.8) | 16 (72.7) | 19 (86.4) | |
| CD4 ≤100 | 77 (69.4) | 34 (44.2) | 42 (53.9) | |
| Clinical Stage | | | | |
| 1 | 19 (17.2) | 12 (63.2) | 16 (84.2) | 0.02 |
| 2 | 13 (11.7) | 8 (61.5) | 9 (69.2) | |
| 3 | 55 (49.5) | 31 (56.4) | 36 (65.5) | |
| 4 | 24 (21.6) | 7 (29.2) | 11 (45.8) | |
| **Cryptococcal symptom/CSF status[b]** | | | | |
| Asymptomatic | 49 (45.8) | 35 (71.4) | 44 (89.8) | <0.0001 |
| Any CM symptom/CSF- | 21 (19.6) | 11 (52.4) | 12 (57.1) | |
| Any CM symptom/No CSF test | 4 (3.7) | 3 (75.0) | 3 (75.0) | |
| Symptomatic/CSF+ | 33 (30.8) | 9 (27.3) | 13 (39.4) | |
| **CM symptom (if CSF+) (n = 33)[c]** | | | | |
| Headache | 28 (84.9) | 8 (28.6) | 12 (42.9) | - |
| Fever | 15 (45.5) | 5 (33.3) | 6 (40.0) | |
| Confusion | 9 (27.3) | 5 (55.6) | 5 (55.6) | |
| Convulsion | 6 (18.2) | 1 (16.7) | 2 (33.3) | |
| Neck stiffness | 16 (48.5) | 4 (25.0) | 7 (43.8) | |
| Multiple symptoms | 24 (72.7) | 8 (33.3) | 11 (45.8) | |
| **Total** | **111** | **58** | **72** | |

[a]Note: One patient who transferred out of a pilot facility prior to 6 months is excluded from the survival calculations and not considered for the denominator for LTFU, known deaths, or 6-month survival

[b]4 patients missing symptom status were not included

[c]symptom data include overlap between patients, significance testing not done

[d]Log-rank test comparing survival at 6 months assuming that LTFU died

[e]Comparing all enrolled as outpatients to all enrolled as inpatients

There was no significant difference in hazard of attrition at six months by age, gender, or site of enrollment in univariable analysis. The final Cox proportional hazards regression model controlled for reason for enrolment, CD4 strata at enrolment, and CM symptom/CSF status.

Detection of cryptococcal antigenemia while admitted as an inpatient (IPD) was significantly associated with increased hazard of attrition compared to OPD patients newly presenting to care (aHR: 2.56, 95% CI: 1.07–6.15). The significant harmful effect of inpatient status held when comparing to all patients with antigenemia enrolled from OPD (regardless of reason for enrollment) (not shown). Concurrent CM at the time of positive whole blood CrAg

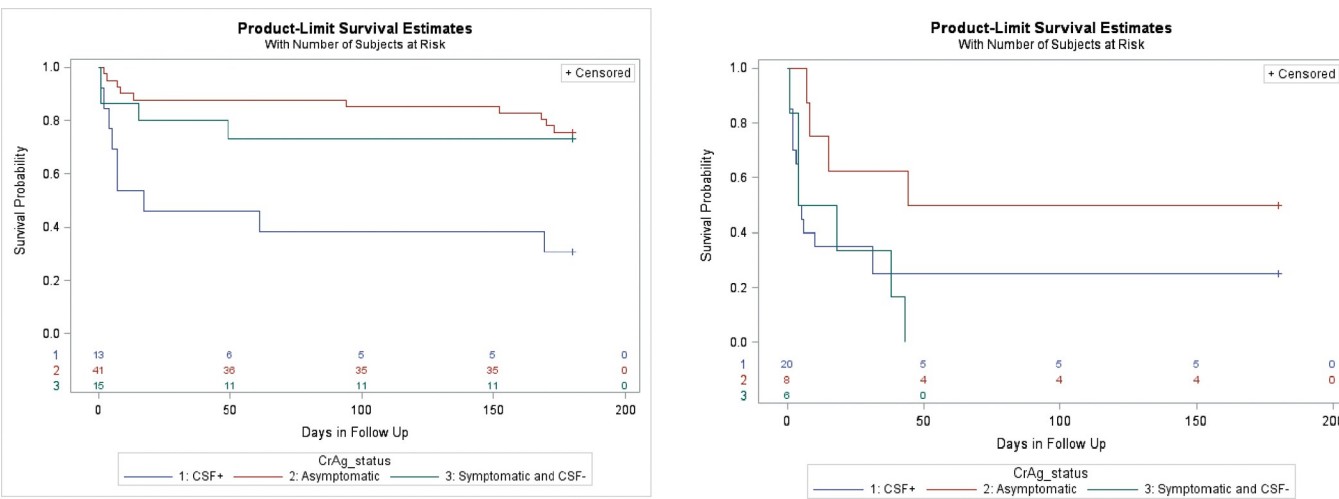

**Fig 1.** Six Month Known Survival of patients with antigenemia by CM symptom/CSF status (assuming LTFU died) in patients enrolled from OPD (a) and IPD (b).

test was also significantly associated with increased hazard of attrition at six months (aHR: 2.48, 95% CI: 1.04–5.92). When excluding patients diagnosed with concurrent CM from the final model, enrollment from IPD was the only significant predictor of attrition (aHR: 6.4, 95% CI: 2.19–18.92) (not shown). When excluding patients with concurrent CM as well as patients enrolled from IPD, no factors were significantly associated with attrition.

For patients with antigenemia enrolled from OPD, concurrent CM was the only significant predictor of attrition at 6 months (aHR: 4.11, 95% CI: 1.56–10.82) (not shown). No variables were significantly associated with the hazard of attrition at 6 months for IPD patients in adjusted analysis. Due to the small number of patients included in these stratified analyses, we present only the results of the combined OPD and IPD model including patients with concurrent CM (**Table 3**).

## Discussion

The observed prevalence of cryptococcal antigenemia of 11.5% and 7.9% for PLHIV with CD4<100 and <200 respectively is higher than that typically reported at these CD4 thresholds, and is more than double the pooled prevalence in patients with CD4<200 reported in a comprehensive literature review [14]. The high prevalence estimate in our study may indicate a higher prevalence of cryptococcal antigenemia in Malawi, compared to countries represented in previous publications. Notably, our prevalence estimate is far higher than that observed (3.5%) in PLHIV with CD4<100 cells/mm$^3$ in a previous single-center pilot of CrAg screening in Malawi [15]. The higher prevalence in our study may possibly reflect an increased yield of patients detected with antigenemia compared to previously reported prevalence from the region, due to Malawi's recent national recommendation to purposefully target CD4 testing to severely ill PLHIV, ART naïve PLHIV, ART-experienced patients with suspected or confirmed treatment failure, and PLHIV who have had interruptions in ART and returned to care. The inclusion of severely ill and symptomatic patients in this approach is likely to lead to higher prevalence estimates. Our results support the effectiveness of this approach in identifying patients at the highest risk of cryptococcal antigenemia. It should be noted that one hospital in this pilot study identified an extremely high prevalence of cryptococcal antigenemia (~27%) and thus contributed substantially to our overall prevalence estimate. The lab staff at this site appropriately performed all

**Table 3. Risk Factors for attrition at six months: Simple and multivariable cox proportional hazards regression.**

| Factor | Cryptococcal antigenemia N = 111[a] (%) | Attrition (died or LTFU) at 6 months N = 53 (%) | Known Survival N = 58 (%) | Bivariable Analysis | | | Multivariable Analysis | | |
|---|---|---|---|---|---|---|---|---|---|
| | | | | HR | 95% Confidence Interval | P-value | aHR | 95% Confidence Interval | P-value |
| | | | | | Lower \| Upper | | | Lower \| Upper | |
| **Age (median, IQR)** | | | | | | | | | |
| | 40 (33–48) | 39.5 (33–47.5) | 41 (33–49) | 1.0 | 0.98 \| 1.03 | 0.97 | - | - \| - | - |
| **Gender** | | | | | | | | | |
| Female | 41 (36.9) | 20 (48.8) | 21 (51.2) | Ref | - \| - | - | - | - \| - | - |
| Male | 70 (63.1) | 33 (47.1) | 37 (52.9) | 0.96 | 0.55 \| 1.67 | 0.88 | - | - \| - | - |
| **Site** | | | | | | | | | |
| Bangwe | 10 (9.0) | 3 (30.0) | 7 (70.0) | Ref | - \| - | - | - | - \| - | - |
| Dedza | 13 (11.7) | 5 (38.5) | 8 (61.5) | 1.27 | 0.30 \| 5.33 | 0.74 | - | - \| - | - |
| Jenda | 33 (29.7) | 13 (39.4) | 20 (60.6) | 1.33 | 0.38 \| 4.68 | 0.65 | - | - \| - | - |
| Mzuzu | 33 (29.7) | 19 (57.6) | 14 (42.4) | 2.48 | 0.73 \| 8.40 | 0.14 | - | - \| - | - |
| Thyolo | 22 (19.8) | 13 (59.1) | 9 (40.9) | 2.85 | 0.81 \| 10.03 | 0.10 | - | - \| - | - |
| **Reason for Enrollment[b]** | | | | | | | | | |
| New HIV | 33 (29.7) | 10 (30.3) | 23 (69.7) | Ref | - \| - | - | Ref | - \| - | - |
| Defaulter | 12 (10.8) | 6 (50.0) | 6 (50.0) | 2.11 | 0.77 \| 5.81 | 0.15 | 1.60 | 0.57 \| 4.50 | 0.37 |
| ART Failure | 27 (24.3) | 7 (25.9) | 20 (74.1) | 0.88 | 0.33 \| 2.31 | 0.79 | 0.85 | 0.32 \| 2.27 | 0.75 |
| Inpatient | 39 (35.1) | 30 (76.9) | 9 (23.1) | 4.37 | 2.12 \| 9.03 | <0.0001 | 2.56 | 1.07 \| 6.15 | 0.04 |
| **CD4** | | | | | | | | | |
| CD4≤100 | 77 (69.4) | 43 (55.8) | 34 (44.2) | Ref | - \| - | - | Ref | - \| - | - |
| CD4 101–199 | 22 (19.8) | 6 (27.3) | 16 (72.7) | 0.37 | 0.16 \| 0.87 | 0.02 | 0.49 | 0.20 \| 1.17 | 0.11 |
| CD4≥200 | 12 (10.8) | 4 (33.3) | 8 (66.7) | 0.46 | 0.16 \| 1.29 | 0.14 | 0.72 | 0.21 \| 2.42 | 0.59 |
| **Clinical Stage** | | | | | | | | | |
| 1 | 19 (17.1) | 7 (36.8) | 12 (63.2) | Ref | - \| - | - | - | - \| - | - |
| 2 | 13 (11.7) | 5 (38.5) | 8 (61.5) | 1.23 | 0.36 \| 3.55 | 0.84 | - | - \| - | - |
| 3 | 55 (49.6) | 24 (43.6) | 31 (56.4) | 1.29 | 0.56 \| 2.99 | 0.55 | - | - \| - | - |
| 4 | 24 (21.6) | 17 (70.8) | 7 (29.2) | 2.88 | 1.19 \| 6.97 | 0.02 | - | - \| - | - |
| **CrAg Clinical Status** | | | | | | | | | |
| Asymptomatic | 49 (47.6) | 14 (28.6) | 35 (71.4) | Ref | - \| - | - | Ref | - \| - | - |
| Symptomatic/ CSF- | 21 (20.4) | 10 (47.6) | 11 (52.4) | 2.04 | 0.91 \| 4.59 | 0.09 | 1.86 | 0.78 \| 4.47 | 0.16 |
| Symptomatic/CSF + | 33 (32.0) | 24 (72.7) | 9 (27.3) | 4.26 | 2.19 \| 8.29 | <0.0001 | 2.48 | 1.04 \| 5.92 | 0.04 |

Note

[a]Excluding one patient which transferred out of pilot site

[b]Reason for enrollment only captures the initial reason why patients were enrolled. There is no overlap between these categories. ART status was only captured for those detected with antigenemia as outpatients. If detected with antigenemia as an inpatient, data were not collected on whether the patient was new HIV, had defaulted, or has suspected treatment failure.

quality assurance measures on a weekly basis and received routine supervision/monitoring visits from the study staff, with no performance issues noted. Therefore, the results from this hospital are unlikely to represent false-positives and may instead indicate that PLHIV in the catchment area of this hospital are at a higher risk of cryptococcal antigenemia.

Our results support the feasibility of appropriate clinical assessment and follow up of patients with cryptococcal antigenemia performed by regular facility staff. In this study, every asymptomatic patient with antigenemia and over 96% of symptomatic but non-CM patients were prescribed pre-emptive fluconazole. Clinical assessment for signs and symptoms of CM in patients with antigenemia was conducted comprehensively in this pilot, with symptom data reported for over 96% of patients with antigenemia. The coverage of lumbar puncture and CSF testing was very high for symptomatic patients (almost 95%). Unfortunately, no asymptomatic patients with antigenemia received CSF tests due to the difficulty in having asymptomatic patients consent to receive an LP. An increasing body of evidence indicates that all patients with cryptococcal antigenemia should ideally receive a lumbar puncture and CSF-CrAg test to rule out sub-clinical CM [16]. In our study, patients with sub-clinical CM would have gone undetected due to the fact that no patients with asymptomatic antigenemia received an LP and CSF test to rule out CM. Our findings indicate a need to better understand the reasons for refusal of LP and to reinforce the critical importance of LPs in messaging and trainings for clinicians and in-patient educational materials in Malawi. The Southern African Clinicians Society guidelines on cryptococcal diagnosis and management provides additional guidance on the need for LP and CSF testing even for asymptomatic patients, and could be drawn upon to inform patient management policy and training efforts for Malawian clinicians in the future [17]. New CrAg LFA tests which determine CrAg status as well as an approximate CrAg titer, which is associated with likelihood of subclinical CM, are currently under evaluation in the region [18]. These tests could potentially provide a method to determine likelihood of subclinical CM and rapidly initiate more intensive therapy for high-CrAg-titer patients when LP is not available or not consented to.

Known survival of patients with antigenemia enrolled from OPD was suboptimal (68%). However, survival in asymptomatic patients enrolled from OPD was at least 76% and was over 95% when assuming that LTFU patients survived (only 2 of 41 known to have died). Therefore, the true survival of asymptomatic patients in our study was very similar to the survival of asymptomatic patients with antigenemia who received pre-emptive fluconazole in a recent clinical trial in Uganda [19]. The fact that no patients with antigenemia without concurrent CM were diagnosed with CM following initiation of pre-emptive fluconazole in our study, and that only 4 patients developed any new symptoms of CM during six months of pre-emptive fluconazole therapy provides evidence that pre-emptive fluconazole and ART for patients with antigenemia without concurrent CM at baseline may have prevented breakthrough CM in some of these patients. This is supported by studies which have shown that the majority of CM occurs soon after ART initiation in patients with antigenemia who are not pre-emptively treated, and almost all occurs by six months of follow up [20, 21]. However, due to LTFU and a lack of comprehensive data on cause of death, we cannot say with certainty that no patients developed CM following enrollment and pre-emptive fluconazole therapy. Our results show that there is also a need to reinforce routine education for clinicians on the importance of performing LP and CSF CrAg-testing in patients showing new symptoms of CM who were previously antigenemic and asymptomatic or CSF-CrAg negative.

Overall attrition (confirmed death or LTFU) at six months of patients with antigenemia was very high, at 47.7%. Even assuming that LTFU patients survived, the overall survival estimate is still low, at around 65%. The high rate of attrition in patients with antigenemia was driven largely by exceedingly poor outcomes for patients with antigenemia detected while admitted to IPD (only 23% known to have survived to six months) and patients diagnosed with concurrent CM (only 27.3% known to have survived to six months). Patients who were detected with antigenemia while admitted to IPD were far more likely to be dead or LTFU at six months, even when adjusting for relevant clinical factors, including concurrent CM. When looking only at patients without concurrent CM, patients enrolled from IPD were more than 6

times as likely to be dead or LTFU at six months. Notable, all 6 IPD patients with symptoms of CM but negative CSF CrAg results died in the first 60 days of follow-up. Although the CSF CrAg LFA has very high sensitivity in detecting cryptococcal antigen in CSF, false negatives have been reported [22]. Because, the Malawian pilot study did not conduct diagnostic tests in addition to the LFA, we cannot rule out the possibility of false negative results. Our results indicate that cryptococcal antigenemia is highly associated with poor outcomes for patients admitted to IPD, and may not be solely due to a higher likelihood of CM in these patients. These results potentially reflect that survival of inpatient PLHIV with AHD may be very low regardless of antigenemia status, due to higher likelihood of these patients having other life-threatening infections or conditions. Unfortunately, this study did not collect data on CrAg-negative inpatients with AHD for comparison.

A substantial number of patients with antigenemia were found to have concurrent CM in this pilot, including ambulatory patients arriving at facility OPD who were ART-experienced (but who had defaulted or had suspected treatment failure). Survival of patients with concurrent CM was very low regardless of the department in which the patients were identified (25% known survival in IPD; 31% known survival in OPD). Treatment for concurrent CM in our pilot study was complicated at some study sites due to the need for patients to reach tertiary referral hospitals for treatment with amphotericin B-containing regimens, and the low survival of patients diagnosed with concurrent CM is unfortunately not surprising due to the small proportion of patients who were confirmed to have received amphotericin B during the induction phase of treatment. A retrospective analysis of CM-related mortality in Botswana found that the one-year mortality rate for patients with CM confirmed to be receiving amphotericin B and fluconazole was still very high, at 65% [23]. Even under clinical trial settings, 10-week mortality for patients receiving 2 weeks of amphotericin B and fluconazole was 45% [24]. This same clinical trial showed that survival is significantly improved when using flucytosine (5-FC) in conjunction with either amphotericin B or fluconazole. Recent clinical trial evidence has shown that 1 day of high dose liposomal amphotericin B with a 2-week fluconazole and flucytosine backbone was non-inferior to one week of amphotericin B and flucytosine, and significantly reduced severe side effects associated with prolonged use of amphotericin B deoxy-cholate [25]. This regimen is now recommended by the World Health Organization as the preferred treatment regimen for HIV-associated CM. No liposomal amphotericin B or flucytosine was available for routine use in Malawi at the time of the study. Fortunately, both of these antifungals have become more available in Malawi since the study period.

Improved access to gold-standard CM drugs, 5-FC and liposomal amphotericin B, as well as intensive training for clinicians on administration and management of these drugs is drastically needed in order to improve the survival of patients with CM in Malawi. In addition to intensive training of clinicians on the use of gold-standard CM treatment regimens, increased availability of these regimens at secondary (District level) hospitals in Malawi could improve outcomes by allowing for more rapid access to treatment for patients with CM.

Overall, the results of this study indicate a need to increase access to routine CrAg screening for patients with advanced HIV, antifungals for treatment of antigenemia, and rapid ART treatment as appropriate, in order to reduce the number of PLHIV who have CM at the time of engaging or re-engaging with ART care. The very poor survival of inpatients with antigenemia indicates a need to identify antigenemia in these patients earlier. PLHIV admitted to IPD should be prioritized for rapid CrAg testing at the time of admission, in addition to rapid diagnostic work-up for other common OIs. Comprehensive CrAg screening and pre-emptive treatment programs centered at OPD/HIV clinics could also reduce the number of PLHIV with antigenemia as a complicating factor of or cause of hospitalization. Lastly, the observation that survival was not-significantly different across the study sites (and therefore all the levels of

healthcare) shows that CrAg screening and pre-emptive treatment may be just as effective at the primary level of care, and should be considered in addition to optimized referral networks to ensure that patients with antigenemia suspected or confirmed to have CM are able to reach hospitals where gold-standard treatment is available.

Limitations of this analysis include a lack of comprehensive antifungal treatment data for some patients diagnosed with CM, as well as a lack of data on ART re-initiation or ART switch for patients detected with antigenemia after ART interruption or suspected ART failure. The lack of complete treatment data for patients with CM limits the conclusions that can be drawn from these results on the efficacy of the CM treatment regimen recommended in Malawi at the time of the pilot. The lack of granular ART regimen data for ART-experienced patients in this analysis means that conclusions cannot be drawn on any impact that detection of crypto-coccal antigenemia and pre-emptive fluconazole may have had on ART switching and/or ART re-initiation. Additionally, due to a small sample size, our analysis could not reliably determine whether delayed ART for ART-naïve outpatients with antigenemia and no concurrent CM was significantly associated with patient outcome.

## Conclusion

To our knowledge, this is the first multi-site/multi-region study to assess the prevalence of cryptococcal antigenemia and the six-month survival and risk factors for attrition of PLHIV with cryptococcal antigenemia in Malawi. CrAg screening of the target patient populations recommended in Malawi's National HIV Guidelines followed by CM symptom assessment and LP/CSF-CrAg testing detected a large proportion of cryptococcal antigenemia and CM in patients with AHD. Our results show that overall known survival of patients with cryptococcal antigenemia was very low, largely due to a large proportion of patients with antigenemia having concurrent CM and/or other severe conditions requiring hospitalization at the time that cryptococcal antigenemia was detected. Survival in asymptomatic patients without CM was relatively high, and no CM was detected in any patients with antigenemia who received pre-emptive fluconazole. Routine CrAg screening, including rapid CrAg testing of PLHIV who are severely ill, has the potential to reduce HIV-related mortality in Malawi through earlier diag-nosis of cryptococcal antigenemia and through improved detection of CM. The results of this study provide valuable evidence on patients who are at high risk of HIV-associated mortality due to CM, and can inform national decision-making as routine CrAg screening and gold-standard CM treatment is scaled up in Malawi in an effort to prevent HIV-associated deaths.

## Supporting information

**S1 Data.**
(XLS)

**S1 File.**
(DOCX)

**S2 File. Inclusivity in global research.**
(DOCX)

## Acknowledgments

The authors gratefully acknowledge the CDC Malawi, Department of Diagnostics (MoH), Department of HIV and AIDS for their technical support in the implementation of this proj-ect, College of Medicine and Mzuzu University for the collaborative support.

## Author Contributions

**Conceptualization:** Master R. O. Chisale, Pocha S. Kamudumuli, Bernard Mvula, James Kandulu, Tsung-Shu Joseph Wu.

**Data curation:** Master R. O. Chisale, James Kandulu, Ben Chilima, Frank W. Sinyiza, Pauline Katundu, Rose Nyirenda, Greg Greene, Tom Chiller.

**Formal analysis:** Alex Jordan.

**Funding acquisition:** Master R. O. Chisale, Pocha S. Kamudumuli.

**Investigation:** Master R. O. Chisale, Alice Maida, Frank W. Sinyiza, Pauline Katundu, Hsin-yi Lee, Rose Nyirenda, Tom Chiller.

**Methodology:** Master R. O. Chisale, Alex Jordan, Pocha S. Kamudumuli, Bernard Mvula, Michael Odo, Alice Maida, James Kandulu, Ben Chilima, Frank W. Sinyiza, Pauline Katundu, Hsin-yi Lee, Tsung-Shu Joseph Wu, Joseph Bitirinyo, Rose Nyirenda, Thoko Kalua, Greg Greene, Tom Chiller.

**Project administration:** Master R. O. Chisale, Alex Jordan, Pocha S. Kamudumuli, Bernard Mvula, Michael Odo, Alice Maida, James Kandulu, Ben Chilima, Frank W. Sinyiza, Pauline Katundu, Hsin-yi Lee, Rebecca Mtegha, Tsung-Shu Joseph Wu, Joseph Bitirinyo, Rose Nyirenda, Thoko Kalua, Greg Greene, Tom Chiller.

**Resources:** Master R. O. Chisale, Pocha S. Kamudumuli.

**Supervision:** Master R. O. Chisale, Alex Jordan, Pocha S. Kamudumuli, Bernard Mvula, Michael Odo, Alice Maida, James Kandulu, Ben Chilima, Frank W. Sinyiza, Pauline Katundu, Hsin-yi Lee, Rebecca Mtegha, Tsung-Shu Joseph Wu, Joseph Bitirinyo, Rose Nyirenda, Thoko Kalua, Greg Greene, Tom Chiller.

**Validation:** Master R. O. Chisale.

**Visualization:** Master R. O. Chisale.

**Writing – original draft:** Master R. O. Chisale, Alex Jordan.

**Writing – review & editing:** Master R. O. Chisale, Alex Jordan, Pocha S. Kamudumuli, Bernard Mvula, Michael Odo, Alice Maida, James Kandulu, Ben Chilima, Frank W. Sinyiza, Pauline Katundu, Hsin-yi Lee, Rebecca Mtegha, Tsung-Shu Joseph Wu, Joseph Bitirinyo, Rose Nyirenda, Thoko Kalua, Greg Greene, Tom Chiller.

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
