## [Decision Letter · Decision Letter 0]

21 Sep 2022

PONE-D-22-17314Six Months Survival and Risk Factors for attrition for patients detected with cryptococcal antigenemia through screening in MalawiPLOS ONE

Dear Dr. Chisale,

Thank you for submitting your manuscript to PLOS ONE. After careful consideration, we feel that it has merit but does not fully meet PLOS ONE’s publication criteria as it currently stands. Therefore, we invite you to submit a revised version of the manuscript that addresses the points raised during the review process.

I will look forward to your detailed responses to the comments of the reviewers in a revised manuscript.

We look forward to receiving your revised manuscript.

Kind regards,

Rodney D Adam

Academic Editor

PLOS ONE

Journal Requirements:

a) Did participants provide their written or verbal informed consent to participate in this study?

b) If consent was verbal, please explain i) why written consent was not obtained, ii) how you documented participant consent, and iii) whether the ethics committees/IRB approved this consent procedure

3. Please include a complete copy of PLOS’ questionnaire on inclusivity in global research in your revised manuscript. Our policy for research in this area aims to improve transparency in the reporting of research performed outside of researchers’ own country or community. The policy applies to researchers who have travelled to a different country to conduct research, research with Indigenous populations or their lands, and research on cultural artefacts. The questionnaire can also be requested at the journal’s discretion for any other submissions, even if these conditions are not met.  Please find more information on the policy and a link to download a blank copy of the questionnaire here: https://journals.plos.org/plosone/s/best-practices-in-research-reporting. Please upload a completed version of your questionnaire as Supporting Information when you resubmit your manuscript

4. You indicated that ethical approval was not necessary for your study. We understand that the framework for ethical oversight requirements for studies of this type may differ depending on the setting and we would appreciate some further clarification regarding your research. Could you please provide further details on why your study is exempt from the need for approval and confirmation from your institutional review board or research ethics committee (e.g., in the form of a letter or email correspondence) that ethics review was not necessary for this study? Please include a copy of the correspondence as an "Other"" file.

"No Conflict of Interest"

"The authors gratefully acknowledge the CDC Foundation (Project 950) for financial support on this project, CDC Malawi, Department of Diagnostics (MoH), Department of HIV and AIDS for their technical support in the implementation of this project, Luke International Malawi for financial management support, College of Medicine and Mzuzu University for the collaborative support. "

"The study was conducted according to the guidelines of the Declaration of Helsinki and approved by the National Health Science Research Committee (NHSRC) of Malawi (Protocol # 18/01/1961). CDC staff were determined to be non-engaged in the pilot screening activities and therefore the pilot was determined to be exempt from Centers for Disease Control and Prevention (CDC) IRB review."

Reviewers' comments:

Reviewer's Responses to Questions

**Comments to the Author**

1. Is the manuscript technically sound, and do the data support the conclusions?

Reviewer #1: Yes

Reviewer #2: Yes

2. Has the statistical analysis been performed appropriately and rigorously? 

Reviewer #1: Yes

Reviewer #2: Yes

3. Have the authors made all data underlying the findings in their manuscript fully available?

Reviewer #1: Yes

Reviewer #2: Yes

4. Is the manuscript presented in an intelligible fashion and written in standard English?

Reviewer #1: Yes

Reviewer #2: Yes

5. Review Comments to the Author

Reviewer #1: This is an interesting and well done paper. I have no major comments to make. The objectives, methodology and conclusions are clear and well presented. A few minor corrections.

- Line 172: Should be CD4 equal to or greater than 200 (rather than CD4 count less than 200)

- Table 1 totals- It is difficult for the reader to keep track of what the denominator is for each value shown

Reviewer #2: This is an observational study of patients screened for CrAgaemia in a pilot, to determine outcome and risk factors for attrition (loss to follow up or death) among CrAg+ PLHIV in Malawi. Patients were screened for CrAg if they had CD4 <200, clinical stage 3 or 4, or admitted to hospital, in line with national guidelines, as part of a pilot study. They found a prevalence of 5.2%, with a wide range from 3.8% to 25.8% *discussed?*. Survival 52.3% - 64.9%, lower survival in concurrent CM. Patients had a higher risk of attrition if inpatient. They conclude that findings indicate CrAg screening is needed to prevent CM, although the findings do not actually relate to this conclusion.

Introduction

2) Line 66-68: Please update the global burden figures to the Rajasingham et al 2022 publication

3) Not clear why this is called a pilot study – the pilot was to introduce screening and assess prevalence, but this study is observational cohort looking at risk of attrition

Methods

- Line 100: again this is an observational cohort study

- It is not clear what the eligibility criteria is. Initial description included ART criteria (line 103), then line 114 mentions a CD4 criteria. Please clarify

Data collection and procedures

- Please specify how CrAg tests were done – finger prick? Pipetted blood? Sample tested following venepuncture? This can affect prevalence: Point of Care Cryptococcal Antigen Screening: Pipetting Finger-Prick Blood Improves Performance of Immuno-Mycologics Lateral Flow Assay - PMC (nih.gov)

- What was the median time between crag test and enrolment to the study/ assessment for symptoms? could this have led to the high proportion of patients with symptoms

Data analysis

- Please also analyse mortality according to ART initiation eg. Immediate vs delayed

Results

The results are hard to follow and read as a list of percentages. I would rather see the total survival/LTFU and then the breakdown could be just in the tables. The text should describe the results of analyses eg. HR by CD4 count, CM status, site analysed first presented if LTFU assumed to have died and then presented if LTFU assumed to have survived.

- Line 179 please add percentage

- Were all asymptomatic patients offered a LP – what number is this?

- Table 1 – please add median CD4, inpatient/outpatient breakdown and proportion with clin stage ¾, perhaps change round columns/rows, this may shed light on the difference in prevalence between sites

- Table 2 – unclear why ART status is only presented for outpatients – please clarify or change to present in both groups

- Line 216 – confusing to switch between survival % to died% please report known to survive

- Please show HR for concurrent CM – this should also be in the abstract

- Please removed CM symptom/no CSF test from the analysis – n=4 meaningless to include

- Remove figure 1b – it isn’t clear why the survival by CM status has been stratified by inpatient/outpatient – was there interaction? Suggest present together as the numbers are too low to be meaningful in the inpatients

- Table 3 – suggest split 'reason for enrolment' to ART status and inpatient vs outpatient - not appropriate to include these in same variable.

- Suggest remove symptomatic if no CSF data – small numbers

Conclusion

- Prevalence may be higher than previously reported as included symptomatic as well as asymptomatic patients? Please add comment.

- Also comment on the mortality difference between asymptomatic and symptomatic with negative LP – could these be CSF negative CM – previously reported by David Boulware group in Uganda, please reference and discuss

- 294-304 repeated points should be removed and should also discuss need for understanding reasons for asymptomatic patients declining LPs

- Line 330 – this HR should be presented in results before commenting on in conclusions

- Line 371 – suggest replace term missingness

- Key message for summary is high prevalence using these screening criteria (which differ from elsewhere), feasible to screen at point of care, with high rates of assessment and treatment received (although LP uptake needs improvement). Survival is higher in CrAgaemic patients screened in outpatients than inpatients, and treatment prevented any known cases of CM in outpatients. Important therefore to screen and initiate treatment before patients become unwell and are admitted, eg. At point of care at the time of HIV diagnosis.

6. PLOS authors have the option to publish the peer review history of their article (what does this mean?). If published, this will include your full peer review and any attached files.

Reviewer #1: No

Reviewer #2: No

---

## [Author Response · Author response to Decision Letter 0]

16 Nov 2022

Reviewers’ comments and Authors Responses 

Reviewer #1: This is an interesting and well-done paper. I have no major comments to make. The objectives, methodology and conclusions are clear and well presented. A few minor corrections.

- Line 172: Should be CD4 equal to or greater than 200 (rather than CD4 count less than 200)

- Table 1 totals- It is difficult for the reader to keep track of what the denominator is for each value shown

Author response: Thank you for the comment. The cut-off used was <200, and so theoretically a CD4 result of 200 would not have received a CrAg test (there were no CD4 results of 200). We have added denominators to Table 1.

Reviewer #2: This is an observational study of patients screened for CrAgaemia in a pilot, to determine outcome and risk factors for attrition (loss to follow up or death) among CrAg+ PLHIV in Malawi. Patients were screened for CrAg if they had CD4 <200, clinical stage 3 or 4, or admitted to hospital, in line with national guidelines, as part of a pilot study. They found a prevalence of 5.2%, with a wide range from 3.8% to 25.8% *discussed?*. Survival 52.3% - 64.9%, lower survival in concurrent CM. Patients had a higher risk of attrition if inpatient. They conclude that findings indicate CrAg screening is needed to prevent CM, although the findings do not actually relate to this conclusion.

Response from Author: Additional text has been added to the discussion and conclusion to reflect the statement that CrAg screening and pre-emptive treatment is needed in order to rapidly identify CM in those presenting to care with CM, and to prevent future onset of CM in patients who are antigenemic.

Introduction

2) Line 66-68: Please update the global burden figures to the Rajasingham et al 2022 publication

Response from author: The updated burden estimates have been added to the background section.

3) Not clear why this is called a pilot study – the pilot was to introduce screening and assess prevalence, but this study is observational cohort looking at risk of attrition

Response from author: Additional text has been added to the methods section to clarify that the observational follow up of CrAg+ patients identified in the pilot is the focus of this manuscript, and that additional results from the pilot (prevalence, risk factors for CrAgemia) will be presented in a separate manuscript.

Methods

- Line 100: again, this is an observational cohort study

Response from author: have edited this line to clarify.

- It is not clear what the eligibility criteria is. Initial description included ART criteria (line 103), then line 114 mentions a CD4 criteria. Please clarify

Response from author: Additional text has been added to this section to clarify that OPD patients were only enrolled if identified with AHD (which was only assessed if patient was ART naïve, interrupted treatment, or suspected/confirmed treatment failure) and that IPD patients were also assessed for AHD but were enrolled regardless of CD4 or clinical stage

Data collection and procedures

- Please specify how CrAg tests were done – finger prick? Pipetted blood? Sample tested following venepuncture? This can affect prevalence: Point of Care Cryptococcal Antigen Screening: Pipetting Finger-Prick Blood Improves Performance of Immuno-Mycologics Lateral Flow Assay - PMC (nih.gov)

Response from author: Clarification on the collection method has been added to the Data Collection section. Testing was done on specimens collected by pipetted fingerprick or by venous collection. Unfortunately, data was not comprehensively collected on the method of collection used for each patient. However, the study team running the pilot are well aware of the implications for sensitivity when using the fingerprick method, and intensive trainings and supervision were carried out to ensure laboratorians were collecting adequate specimens.

- What was the median time between crag test and enrolment to the study/ assessment for symptoms? could this have led to the high proportion of patients with symptoms

Response from author: Per the study protocol, enrollment (including CD4 test, staging, CrAg test, and clinical assessment and treatment) were to be conducted at the point of care on the same day whenever possible. Although comprehensive data was not collected on the time from positive test to symptom assessment for CrAg positive patients, all of CrAg positive patients received symptom assessments on the same day as their positive CrAg test. So, we believe delays in assessment cannot be attributed to the high proportion displaying CM symptoms.

Data analysis

- Please also analyse mortality according to ART initiation eg. Immediate vs delayed

Response from author: additional results have been added for the ART-naïve patients based on ART delay vs immediate. No significant difference in outcomes was observed. Have added mention of this to results, as well as mention that the sample size for the sub-analysis is too small to give substantial evidence.

Results

The results are hard to follow and read as a list of percentages. I would rather see the total survival/LTFU and then the breakdown could be just in the tables. The text should describe the results of analyses eg. HR by CD4 count, CM status, site analyzed first presented if LTFU assumed to have died and then presented if LTFU assumed to have survived.

Response from author: The results section has been edited and re-arranged based on this and other reviewer comments.

- Line 179 please add percentage

Response from author: Have added the percentage

- Were all asymptomatic patients offered a LP – what number is this?

Response from author: As mentioned in the discussion, clinicians were urged to give asymptomatic CrAg-positive patients LPs to rule out sub-clinical CM. However, no asymptomatic patients consented to receive LPs in the pilot.

- Table 1 – please add median CD4, inpatient/outpatient breakdown and proportion with clin stage ¾, perhaps change round columns/rows, this may shed light on the difference in prevalence between sites

Response from author: These additional results are being prepared in a separate manuscript, which will go into much more detail on the prevalence estimates across sites, including a risk factor analysis for AHD and antigenemia across all sites.

- Table 2 – unclear why ART status is only presented for outpatients – please clarify or change to present in both groups

Response from author: Due to limitations in data collection (using routine facility staff), ART data was only comprehensively collected for ART naïve patients enrolled from OPD. The pilot prioritized outpatients who were ART naïve, as this was the only population for whom CrAg status would potentially impact ART treatment decisions. ART data was not collected systematically from inpatients with HIV. 

- Line 216 – confusing to switch between survival % to died% please report known to survive

Response from author: the suggested change has been made

- Please show HR for concurrent CM – this should also be in the abstract

Response from the author: The unadjusted HR for concurrent CM is presented in table 3. We have added the adjusted HR for concurrent CM to the abstract

- Please remove CM symptom/no CSF test from the analysis – n=4 meaningless to include

Response from the author: Have removed from the analysis per your suggestion.

- Remove figure 1b – it isn’t clear why the survival by CM status has been stratified by inpatient/outpatient – was there interaction? Suggest present together as the numbers are too low to be meaningful in the inpatients

Response from authors: Kaplan Meir analysis showed that survival differed significantly by symptom/CSF status in outpatients, but not in inpatients. And the survival of all clinical categories was far lower in IPD. Therefore, we felt that combining these risks conflating possible unmeasured effects of severe illness in IPD CrAg+ patients, and by stratifying we give a clearer picture of survival. 

- Table 3 – suggest split 'reason for enrolment' to ART status and inpatient vs outpatient - not appropriate to include these in same variable.

Response from author: We do not have comprehensive ART data for inpatients, and so inpatients are a category of their own, agnostic to the ART criteria (which are only for OPD patients). Because there is no overlap between these groups, we have left it unchanged.

- Suggest remove symptomatic if no CSF data – small numbers

Response from author: We have removed these from the analysis

Conclusion

- Prevalence may be higher than previously reported as included symptomatic as well as asymptomatic patients? Please add comment.

Response from author: Have specifically added this into the discussion text.

- Also comment on the mortality difference between asymptomatic and symptomatic with negative LP – could these be CSF negative CM – previously reported by David Boulware group in Uganda, please reference and discuss

Response from author: We have added mention of this possibility, along with the reference 

- 294-304 repeated points should be removed and should also discuss need for understanding reasons for asymptomatic patients declining LPs

Response from author: additional text has been added iterating the need to gain a better understanding of LP refusal

- Line 330 – this HR should be presented in results before commenting on in conclusions

Response from authors: This HR is mentioned in the results.

- Line 371 – suggest replace term missingness

 Response from authors: Have replaced

- Key message for summary is high prevalence using these screening criteria (which differ from elsewhere), feasible to screen at point of care, with high rates of assessment and treatment received (although LP uptake needs improvement). Survival is higher in CrAgaemic patients screened in outpatients than inpatients, and treatment prevented any known cases of CM in outpatients. Important therefore to screen and initiate treatment before patients become unwell and are admitted, eg. At point of care at the time of HIV diagnosis.

Response from author: The discussion section has been updated to more clearly capture the suggested key points.

---

## [Decision Letter · Decision Letter 1]

13 Jan 2023

PONE-D-22-17314R1Six Months Survival and Risk Factors for attrition for patients detected with cryptococcal antigenemia through screening in MalawiPLOS ONE

Dear Dr. Chisale,

Thank you for submitting your manuscript to PLOS ONE. After careful consideration, we feel that it has merit but does not fully meet PLOS ONE’s publication criteria as it currently stands. Therefore, we invite you to submit a revised version of the manuscript that addresses the points raised during the review process.

I look forward to receiving a revision that addresses the points raised by the reviewer.

We look forward to receiving your revised manuscript.

Kind regards,

Rodney Adam

Academic Editor

PLOS ONE

Journal Requirements:

Reviewers' comments:

Reviewer's Responses to Questions

**Comments to the Author**

1. If the authors have adequately addressed your comments raised in a previous round of review and you feel that this manuscript is now acceptable for publication, you may indicate that here to bypass the “Comments to the Author” section, enter your conflict of interest statement in the “Confidential to Editor” section, and submit your "Accept" recommendation.

Reviewer #2: (No Response)

2. Is the manuscript technically sound, and do the data support the conclusions?

Reviewer #2: Partly

3. Has the statistical analysis been performed appropriately and rigorously? 

Reviewer #2: Yes

4. Have the authors made all data underlying the findings in their manuscript fully available?

Reviewer #2: No

5. Is the manuscript presented in an intelligible fashion and written in standard English?

Reviewer #2: (No Response)

6. Review Comments to the Author

Reviewer #2: The manuscript reports important findings related to cryptococcal antigen screening and has been amended in line with previous review. There are some minor remaining issues that would be beneficial to address.

Study design

- Line 97 Cross-sectional pilot is a confusing term – suggest remove cross-sectional.

- Line 154 should read mortality not morbidity

Results

- A Cox regression model has been used to identify variables associated with mortality.

- In this wide-ranging cohort, the variables associated with mortality are already known/quite obvious e.g. inpatients would be expected to have a greater mortality than outpatients, same with lower CD4 and clinical stage. It is not clear what the value of this analysis is particularly since there is no CrAg negative group – we don’t know whether these mortality associations are related to cryptococcus, likely not.

- What is most interesting from the results is that symptomatic/CSF negative patients are more likely to die than asymptomatic

- Suggest reworking results section to focus on survival analysis by Symptom/CSF status, adjusting for variables that are known to be associated with mortality and only stratifying by inpatient/outpatient status if there is an interaction

- Suggest relabel figures to ‘Kaplan-Meier’

- Table 3 – suggest making clear which categories in ‘reason for enrolment’ are outpatients – it is not clear if a patient could be in both ‘new HIV’ and ‘inpatient’ categories

Discussion

- The most interesting finding of the association of mortality with symptoms is not discussed. 6 IPD with symptoms but CSF negative are mentioned as false negatives but 21 patients fit this description. The fact that mortality is greater in this group than the asymptomatic group indicates that despite negative LP, they may have CNS cryptococcus (or alternatively another undiagnosed CNS pathology). The negative CSF may be because of the ‘hook effect’ – suggest make clear if CSF was tested for this e.g. following dilution, or due to ‘CSF -negative CM’, which has previously been described. Did any have scans to exclude cryptococcomas?

- 318 – suggest sticking to %attrition or %survival throughout rather than switching between – since many were LTFU and survival unknown would be better to refer to %attrition throughout this paper and this would be more consistent with other literature.

I cannot see the supplementary data or any metadata although it does say that all data will be available without restriction.

7. PLOS authors have the option to publish the peer review history of their article (what does this mean?). If published, this will include your full peer review and any attached files.

Reviewer #2: No

---

## [Author Response · Author response to Decision Letter 1]

2 Mar 2023

Reviewers’ comments and Authors Responses 

Study design

- Line 97 Cross-sectional pilot is a confusing term – suggest remove cross-sectional.

Author response: This term has been removed.

- Line 154 should read mortality not morbidity

Author response: This suggested edit has been made

Results

- A Cox regression model has been used to identify variables associated with mortality.

- In this wide-ranging cohort, the variables associated with mortality are already known/quite obvious e.g. inpatients would be expected to have a greater mortality than outpatients, same with lower CD4 and clinical stage. It is not clear what the value of this analysis is particularly since there is no CrAg negative group – we don’t know whether these mortality associations are related to cryptococcus, likely not.

Author response: While the authors agree that there are no major surprises in the risk factors for attrition which were identified, the authors believe this analysis still provides value, as there is very little existing data on outcomes/survival PLWHV detected with antigenemia through routine CrAg screening in the context of near-standard of care implementation in Sub-Saharan Africa, and almost no data from Malawi outside the context of randomized control trials. Additionally, our results on 6 month mortality of patients with concurrent CM under near-standard of care provide valuable evidence for the Malawian HIV program which illustrates the dire outcomes associated with this disease in settings which more closely reflect the reality of programmatic care in Malawi (as opposed to results of clinical trials in which even the control groups receive “standard of care” which is higher quality than the reality under routine circumstances.

Regarding the second point, the reviewer’s comment is correct in that we cannot definitively state that cryptococcal disease was the direct cause of death in any of these patients. However, we disagree that it is unlikely that cryptococcal disease contributed to death in the patients who died. Per WHO guidelines, a positive blood CrAg result from a patient with advanced HIV disease (and without previously treated disease) indicates a high likelihood of concurrent cryptococcal disease and high risk of progression to more severe forms of invasive disease (including cryptococcal meningitis). A substantial body of scientific literature has demonstrated the extreme impact of cryptococcal disease on mortality, especially in lower resource settings. Therefore we feel it is highly likely that this opportunistic infection (even in those without concurrent CM) contributed to poor outcomes. 

- What is most interesting from the results is that symptomatic/CSF negative patients are more likely to die than asymptomatic

- Suggest reworking results section to focus on survival analysis by Symptom/CSF status, adjusting for variables that are known to be associated with mortality and only stratifying by inpatient/outpatient status if there is an interaction

Response from authors: The results of the manuscript already include both unadjusted and unadjusted survival results for CrAg-positive patients based on symptom/CSF status. For Figure 1, we display the unadjusted survival by symptom/CSF status in outpatients separately from inpatients because the symptom/CSF status was not associated with survival in inpatients, but was for outpatients. Symptom/CSF status is also controlled for in the adjusted Cox Proportional Hazards model which also adjusts for other relevant variables (inclusion/exclusion determined through backward model selection) for which data were collected, and includes both outpatients and inpatients. While there is mention of how the Cox Proportional Hazard model results shifted when stratifying the model by inpatient vs. outpatient, that is not the primary result that we report (i.e. Table 3 is not stratified by inpatient/outpatient). We also report the results of the model when excluding certain subsets of patients logically expected to be at higher risk of attrition (i.e. inpatients, those with concurrent CM).

- Suggest relabel figures to ‘Kaplan-Meier’

Response from authors: This edit has been made

- Table 3 – suggest making clear which categories in ‘reason for enrolment’ are outpatients – it is not clear if a patient could be in both ‘new HIV’ and ‘inpatient’ categories

Response from authors: The reason for enrollment was recorded as ‘inpatient’ for any patient that was detected with antigenemia while admitted to inpatient wards. The other ‘reasons for enrollment’ are only for patients detected with antigenemia while outpatients. A footnote has been added to Table 3 to clarify this.

Discussion

- The most interesting finding of the association of mortality with symptoms is not discussed. 6 IPD with symptoms but CSF negative are mentioned as false negatives but 21 patients fit this description. The fact that mortality is greater in this group than the asymptomatic group indicates that despite negative LP, they may have CNS cryptococcus (or alternatively another undiagnosed CNS pathology). The negative CSF may be because of the ‘hook effect’ – suggest make clear if CSF was tested for this e.g. following dilution, or due to ‘CSF -negative CM’, which has previously been described. Did any have scans to exclude cryptococcomas?

Response from authors: The discussion section currently does not devote much space to discussing the effect of being ‘symptomatic/CSF-‘ on attrition, as this was not significantly associated with attrition in adjusted analysis. This indicates that when controlling for other relevant variables selected based on backward selection, being symptomatic but not CSF CrAg-positive was not predictive of attrition. 

There is no reference in the manuscript to false negatives. A negative CSF CrAg test in a patient with a positive blood CrAg test is not a false negative. A positive blood CrAg result indicates antigenemia, and can often indicate otherwise undetected cryptococcal disease, and can also precede CNS involvement and cryptococcal meningitis, hence the WHO recommendation for CrAg screening and pre-emptive fluconazole to prevent further dissemination and cryptococcal meningitis for those patients who are not CSF-CrAg positive at the time of the positive blood CrAg result.

All follow up for patients detected with positive CrAg results was per Malawian national recommendations, and data was not collected on subsequent diagnostic procedures outside the scope of CrAg testing. While lab staff conducting CrAg testing were trained on how to conduct serial dilution and perform subsequent CrAg testing on dilutions in case of suspected false negative, we unfortunately do not have data on whether any dilution was done prior to positive CrAg result.

- 318 – suggest sticking to %attrition or %survival throughout rather than switching between – since many were LTFU and survival unknown would be better to refer to %attrition throughout this paper and this would be more consistent with other literature.

Response from authors: The authors appreciate this comment, and have debated whether to make the suggested change. However, the current manuscript reflects phrasing which was suggested by previous reviewers, and which the authors feel allows the reader to better understand the large relative impact of LTFU on the overall attrition. Despite this making for a somewhat more tedious reading of the results, it also allows the discerning reader to see the range of potential mortality by describing certain results sections stratified by known mortality and mortality plus LTFU (attrition).

I cannot see the supplementary data or any metadata although it does say that all data will be available without restriction.

Response from authors: The data set will be provided as a supplementary file

---

## [Editor Report · Decision Letter 2]

27 Mar 2023

PONE-D-22-17314R2Six Months Survival and Risk Factors for attrition for patients detected with cryptococcal antigenemia through screening in MalawiPLOS ONE

Dear Dr. Chisale,

Thank you for submitting your manuscript to PLOS ONE. After careful consideration, we feel that it has merit but does not fully meet PLOS ONE’s publication criteria as it currently stands. Therefore, we invite you to submit a revised version of the manuscript that addresses the points raised during the review process.

I have noted that in your revision, line 276 has "data was" rather than "data were". Please note that datum is singular and data plural, so it should be "data were". Please correct this so the manuscript can be accepted.

We look forward to receiving your revised manuscript.

Kind regards,

Rodney Adam

Academic Editor

PLOS ONE
---

## [Author Response · Author response to Decision Letter 2]

28 Mar 2023

Author response: This term has been removed and an appropriately replaced as per your suggestion. Please check on line number 276.

---

## [Editor Report · Decision Letter 3]

29 Mar 2023

Six Months Survival and Risk Factors for attrition for patients detected with cryptococcal antigenemia through screening in Malawi

PONE-D-22-17314R3

Dear Dr. Chisale,

We’re pleased to inform you that your manuscript has been judged scientifically suitable for publication and will be formally accepted for publication once it meets all outstanding technical requirements.

Kind regards,

Rodney Adam

Academic Editor

PLOS ONE
---

## [Editor Report · Acceptance letter]

20 Apr 2023

PONE-D-22-17314R3 

Six months survival and risk factors for attrition for patients detected with cryptococcal antigenemia through screening in Malawi 

Dear Dr. Chisale:

I'm pleased to inform you that your manuscript has been deemed suitable for publication in PLOS ONE. Congratulations! Your manuscript is now with our production department. 

Kind regards, 

on behalf of

Dr. Rodney Adam 

Academic Editor

PLOS ONE